# High Speed and Accuracy of Animation 3D Pose Recognition Based on an Improved Deep Convolution Neural Network

**Wei Ding** [1,*] and **Wenfa Li** [2]

1 School of Journalism & Communication, Anhui Normal University, Wuhu 241002, China
2 The Institute of Artificial Intelligence, University of Science and Technology Beijing, Beijing 100083, China
* Correspondence: dingwei0409@ahnu.edu.cn

**Abstract:** Pose recognition in character animations is an important avenue of research in computer graphics. However, the current use of traditional artificial intelligence algorithms to recognize animation gestures faces hurdles such as low accuracy and speed. Therefore, to overcome the above problems, this paper proposes a real-time 3D pose recognition system, which includes both facial and body poses, based on deep convolutional neural networks and further designs a single-purpose 3D pose estimation system. First, we transformed the human pose extracted from the input image to an abstract pose data structure. Subsequently, we generated the required character animation at runtime based on the transformed dataset. This challenges the conventional concept of monocular 3D pose estimation, which is extremely difficult to achieve. It can also achieve real-time running speed at a resolution of 384 fps. The proposed method was used to identify multiple-character animation using multiple datasets (Microsoft COCO 2014, CMU Panoptic, Human3.6M, and JTA). The results indicated that the improved algorithm improved the recognition accuracy and performance by approximately 3.5% and 8–10 times, respectively, which is significantly superior to other classic algorithms. Furthermore, we tested the proposed system on multiple pose-recognition datasets. The 3D attitude estimation system speed can reach 24 fps with an error of 100 mm, which is considerably less than that of the 2D attitude estimation system with a speed of 60 fps. The pose recognition based on deep learning proposed in this study yielded surprisingly superior performance, proving that the use of deep-learning technology for image recognition has great potential.

**Keywords:** deep convolutional neural network (DCNN); pose recognition; character animation; complex posture; computer graphics

## 1. Introduction

Human motion recognition refers to the recognition and categorization of individuals' behavioral activities using video clips using computer algorithms [1,2]. It is an important avenue of research in computer vision. The keys to pose recognition and two-dimensional (2D) animation generation of characters lie in pose extraction, image compression, and the subsequent extraction and application of each action of the character itself. Convolutional neural networks (CNN) have shown great potential for medical image processing [3,4].

Several film companies use sensor-based schemes [5]. There are two primary methods for three-dimensional (3D) pose recognition. The first is to estimate the 3D pose based on 2D pose recognition. This primarily depends on high-precision 2D attitude recognition results [6]. For example, Simonyan et al. (2014) proposed a scheme based on a feedforward neural network that uses neural networks to directly estimate the corresponding 3D pose from the extracted 2D pose [7]. Based on similar schemes, Facebook AI introduced temporal series information to further improve the pose performance [8], which directly uses images as inputs. Ji et al. (2022) proposed the encoder–decoder network (EDN) model based on a given image [9]. This network was used to predict various parameters of the

parameterized model, thereby predicting the posture of the human body [10]. This model can simultaneously predict the posture and shape of the human body.

Researchers are currently attempting to use a single ordinary RGB camera to estimate human body posture [11]. Traditional square measures are usually supported by artificially designed rules and options such as skin color options. However, this feature is generally unsuitable for scene options such as light [12]. Therefore, similar techniques are typically used for straightforward following, for example, by following the hand. Therefore, several studies have reviewed the use of deep convolutional neural networks (DCNN) to estimate the human body [13–16].

When using feedforward neural networks to generate real-time character animation based on pre-extracted 3D posture data, the neural network is trained using the environmental characteristics and behavioral orientation data near the current character as input and the matching posture is extracted as the expected output [17]. Subsequently, at runtime, the options of the nearby virtual surroundings area unit are continuously extracted and used as input variables in the neural network to create a sound reproduction of character animation. Traditional object recognition techniques accomplish object recognition by matching a set of manually selected feature rules [18–20]. For example, "hand" objects in digital images are recognized through the color of the skin and the shape of the hand. These recognition techniques are limited by feature rules that can be summarized manually [21]. Compiling rule sets is challenging for objects with complex rules. Among various neural network technologies, the feedforward neural network is the simplest. Its core comprises a series of neurons arranged in layers; however, the neurons cannot be freely connected [22]. By contrast, each layer of neurons connects only to the previous layer of neurons, and the output of the previous layer of neurons passes to the next layer of neurons after passing through the activation function [23]. The biggest characteristic of "feedforward" neural networks is that neurons at the same level cannot provide feedback to each other. Therefore, the entire feedforward neural network can be expressed as a directed acyclic graph without a feedback path. Feedforward constraints facilitate implementation, optimization, and learning. Therefore, feedforward neural networks are among the most mature and widely used neural networks [24–27]. However, when processing image data, traditional feedforward neural networks require a large number of neurons to read the pixel information of the input image, making them unsuitable for processing high-pixel images.

The posture dataset enables the detection of targets, postures, semantic segmentation, and strength segmentation. Recently, several posture datasets have been used for experimental comparisons. The Microsoft COCO 2014 dataset was constructed by Lin et al. (2014) [28] and is a set of 2D image datasets provided by Microsoft. It is widely used for tasks such as object detection, pose detection, semantic segmentation, and strength segmentation. It contains keypoint information about the human body in various scenarios. For the COCO pose detection dataset, there were 120,000 sample images, including approximately 64,000 sample images of the human body. For these images, the positions of each keypoint of the human body (2D coordinates) and the visible and invisible (represented by 0, 1, and 2 for unlabeled, invisible, and visible situations, respectively) were manually annotated. The CMU Panoptic dataset is also used as a typical case study [28]. Joo et al. (2015) [29] constructed a spherical dome area where 480 VGA cameras were placed, each with a resolution of 640 × 480 pixels, and they were synchronized using a hardware clock. The captured frame rate was 25 fps. Simultaneously, 31 high-definition cameras were installed to record images at a resolution of 1920 × 1080 pixels, operating at 30 fps while ensuring time synchronization with the VGA cameras. Ten Kinect v2 cameras were used to capture the 3D poses of the human body. The Kinect V2 camera has a resolution of 1920 × 1080 pixels (RGB) and a depth of 512 × 424 pixels (depth). The KinectV2 camera was operated at 30 fps, and they ensured that the time was the same for all the three types of cameras. The source of JTA data collection was Rockstar's Grand Theft Auto 5 [29]. It adopts a physics-based rendering architecture that produces high-precision character animations. It contains a large number of characters with different postures and costumes

for use, as well as a large number of scenes, and provides various weather conditions. Human3.6M [30] is currently recognized as the largest 3D human posture dataset, which includes 15 daily life actions, such as walking, eating, sitting, and walking a dog. The actions were performed by 11 professional actors wearing motion capture equipment in a laboratory environment and were recorded using four synchronous cameras at 50 Hz, which generated approximately 3.6 million images. Data from the seven actors included 3D annotations.

Therefore, this study proposes a series of technical solutions for animation posture recognition and outputs based on neural network structures that exhibit better performance than traditional methods for animation posture output. To overcome the above shortcomings in human posture recognition, this paper proposes the following three improvements in object recognition, 3D posture estimation, and animation generation. (1) Using the concept of "convolution" in signals and systems, a CNN is proposed. CNNs can efficiently process large images by allowing their own neurons to output only surrounding cells within the coverage range. A CNN generally consists of two parts; the first half is composed of multiple convolutional layers, and the second half is composed of fully connected neurons. The parameters in a CNN, called "convolutional kernels", need to be learned one by one in the convolutional layer CNN, and, hence, the amount of parameters that need to be considered is less than that of traditional neural networks. This facilitates the analysis of complex information with fewer parameters. Therefore, CNNs have become an attractive deep learning neural network structure for computer vision. (2) The scheme of directly using images as inputs does not rely on 2D posture data, but directly uses image information to complete 3D posture estimation. This scheme can utilize the light and shadow features of the image to aid in estimation, and, therefore, can provide more accurate results than the previous scheme. However, this method is more challenging and the system complexity is higher. (3) The researcher attempts to generate real-time character animation that conforms to the characteristics of the surrounding environment and character behavior based on the collected 3D posture information, using the surrounding environment and character behavior as input. This method can significantly reduce the workload of animation creators and simplify the complexity of creating high-fidelity virtual character animations. The traditional algorithm adopts the form of a violent search that quickly searches the existing 3D posture database and selects the closest match from the posture data for the output. Such a scheme relies on a large amount of memory and heavily affects CPU performance.

The main contributions of this study are as follows: (1) using a multitasking design method, 2D and 3D poses were successfully recognized in the same network, greatly improving the information transmission and acceleration space of the network; (2) the acceleration scheme used for training as well as the network and runtime visualization systems used for debugging are described, so that the process discussed in this article has not only theoretical value but also practical significance in engineering; (3) the system described in this paper has both data acquisition and post-data acquisition applications, demonstrating the various roles that deep-learning technology can play in the field of computer graphics animation.

The remainder of this paper is organized as follows. Section 2 introduces a 3D pose-estimation algorithm based on deep learning, and Section 3 proposes a 3D pose estimation algorithm for a single camera. Because the 3D pose of a single camera often has multiple solutions, the algorithm in this case generally provides a reasonable 3D pose result; therefore, it is called an estimation algorithm. Based on 3D gesture recognition and estimation, real-time character animation prediction was performed based on user input and trajectory.

## 2. Methodology

### 2.1. 3D Pose Recognition Based on Deep Learning

A new multitasking and multilevel angle-estimation neural network and its corresponding illustration technique were planned. RGB pictures were used as input for multitasking, and the output was 2D joint positions, depths, and affiliations within the

same process [27]. In addition, the multilayer structure avoided repetitive processes and maintained accuracy. The postprocessing system then generated all the human postures that support this information. We propose a technique for representing joints as a bridge between neural networks and ancient algorithms that is less complicated for learning and postprocessing. While the existing visual datasets train neural networks and overcome the shortcomings of 3D knowledge sets in wild house mouse models, we offer a hybrid coaching style that features real 2D and virtual 3D knowledge sets captured from sports. Finally, we validated the effectiveness of our method by exploiting virtual datasets to coach real-world tasks. We offer additional lightweight neural network styles to attain time-period speed. Our postprocessing system overcomes the issue of reduced accuracy.

The 3D pose estimation algorithm framework described in this article is shown in Figure 1. For a given image input, a CNN is first used for processing. The three branches of the CNN will produce three different outputs, each encoding a type of information used to reconstruct a 3D pose. Subsequently, the rapid post processing process uses these three types of information as input to reconstruct the final 3D pose [28].

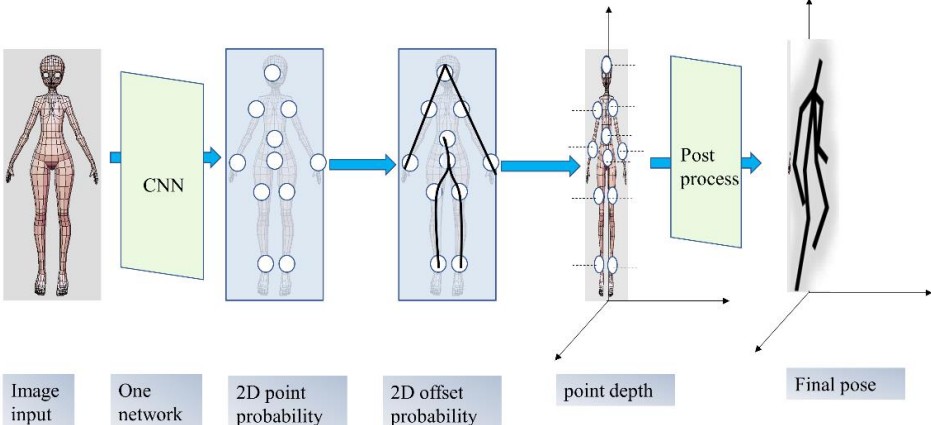

**Figure 1.** Basic framework of 3D attitude estimation algorithms.

In this directed graph, every node corresponds to the joint purpose *j* of an individual's body. Each joint point contains the following information:

$$(Class_j, X_j, Y_j, D_j, OffsetX_j, OffsetY_j) \tag{1}$$

where $Class_j$ represents the current joint point, $X_j, Y_j$ represents the 2D coordinates of the current joint point, $D_j$ represents the distance between the current joint point and the camera, and $OffsetX_j, OffsetY_j$ represents the relative offset. This coding method continues our research in 2D attitude recognition systems. We encoded the 3D joint position in two parts. The 2D position in image space $X_j$, and the relative depth in world space $D_j$ (Figure 2).

Postprocess execution mapping *F* converts the 2D limb joint points into 3D areas as follows:

$$F(Xj, Yj, D) \rightarrow (X_{3D}, Y_{3D}, Z_{3D}) \tag{2}$$

We selected the relative depth of the world space to maximize the mean of the probability distribution of the depth value of the joint points to 0, thereby facilitating the learning of the neural networks. This value is obtained as follows. For each joint, the connecting joint preparing for the highest level becomes the parent node, and the distant joint becomes the child node. Subsequently, the relative depth is encoded because of the excellence between the depth of the current joint and that of the parent joint.

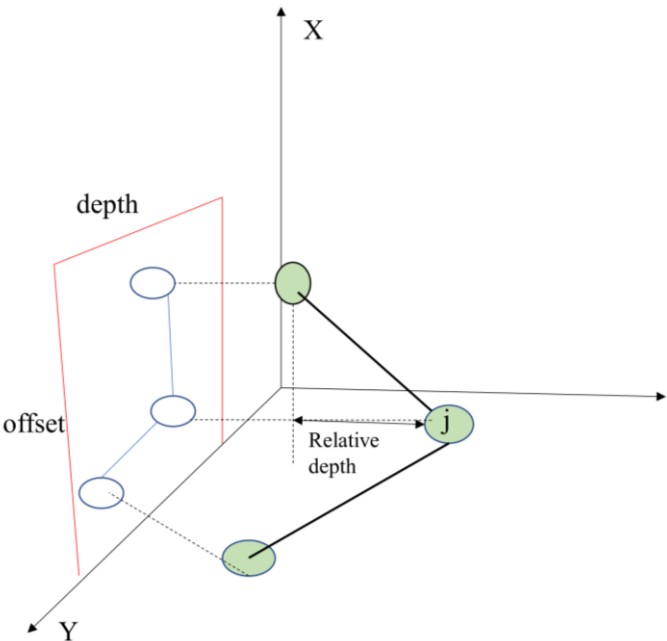

**Figure 2.** Coding of a 3D joint point j.

Obviously, the representation of 3D poses was not unique:

$$\begin{bmatrix} X_1 & Y_1 & Z_1 \\ \dots & \dots & \dots \\ X_n & Y_n & Z_n \end{bmatrix} \tag{3}$$

Each row represents the 3D coordinates of a joint purpose and *n* is the number of joint points of this character. This matrix illustration is employed by networks such as an easy 3D baseline. However, this illustration indicates that network F will perform the subsequent mapping:

$$F\left(\begin{bmatrix} U_1 & V_1 \\ \dots & \dots \\ U_n & V_n \end{bmatrix}\right) = \begin{bmatrix} X_1 & Y_1 & Z_1 \\ \dots & \dots & \dots \\ X_n & Y_n & Z_n \end{bmatrix} \tag{4}$$

Our expected output is as follows:

$$F(image) = \left\{ P_1 \begin{bmatrix} X_1 & Y_1 & Z_1 \\ \dots & \dots & \dots \\ X_n & Y_n & Z_n \end{bmatrix}, P_2, \dots, P_n \begin{bmatrix} X_1 & Y_1 & Z_1 \\ \dots & \dots & \dots \\ X_n & Y_n & Z_n \end{bmatrix} \right\} \tag{5}$$

Therefore, it is difficult to model victimization neural networks. A possible solution is to run network *F* multiple times to output all sequences, as follows:

$$F(image, \{P_1, P_2, \dots, P_{n-1}\}) = P_n \begin{bmatrix} X_1 & Y_1 & Z_1 \\ \dots & \dots & \dots \\ X_n & Y_n & Z_n \end{bmatrix} \tag{6}$$

Another method is to divide the regions of every physical structure and map them individually. We adopt the following distinct approach.

$$F(image) = F(Map_{class}, Map_{offset}, Map_{depth}) \tag{7}$$

Subsequently, we output the results as a series of variable length:

$$C(\{Map_{class}, Map_{offset}, Map_{depth}\})$$
$$= \left\{ P_1 \begin{bmatrix} X_1 & Y_1 & Z_1 \\ \cdots & \cdots & \cdots \\ X_n & Y_n & Z_n \end{bmatrix}, P_2 \begin{bmatrix} X_1 & Y_1 & Z_1 \\ \cdots & \cdots & \cdots \\ X_n & Y_n & Z_n \end{bmatrix}, \dots, P_n \begin{bmatrix} X_1 & Y_1 & Z_1 \\ \cdots & \cdots & \cdots \\ X_n & Y_n & Z_n \end{bmatrix} \right\} \tag{8}$$

We used a multilayer design that supports a feature pyramid network to discover multiple ranges through the network structure. This structure can adapt to the size of the target, but does not unduly increase the method value. As a result of the unit of measurement of the method value savings being redundant, this information is usually saved, but does not decrease accuracy. For each region inside the input image, there is only one optimum detection level, which suggests that various detection levels must not be calculated.

At the 2015 ILSVRC competition, a Residual Network (ResNet) achieved good results [31]. It refines the image feature maps with fewer parameters and a deeper network structure, resulting in image feature representations with better information representation capabilities. We used ResNet34 because the face compressed the channel in 0.5. The detailed structures are listed in Table 1.

**Table 1.** A skeleton network structure table similar to ResNet.

| Layer Name | Output | 34 Layer |
|---|---|---|
| Input layer | $384 \times 384$ | |
| Conv1 | $192 \times 192$ | $7 \times 7, 8$, stride 2 |
| | | $3 \times 3$ max pool, stride 2 |
| Conv2x | $96 \times 96$ | $\begin{bmatrix} 3 \times 3 & 32 \\ 3 \times 3 & 32 \end{bmatrix} \times 3$ |
| Conv3x | $48 \times 48$ | $\begin{bmatrix} 3 \times 3 & 64 \\ 3 \times 3 & 64 \end{bmatrix} \times 4$ |
| Conv4x | $24 \times 24$ | $\begin{bmatrix} 3 \times 3 & 128 \\ 3 \times 3 & 128 \end{bmatrix} \times 6$ |
| Conv5x | $12 \times 12$ | $\begin{bmatrix} 3 \times 3 & 256 \\ 3 \times 3 & 256 \end{bmatrix} \times 3$ |

To reduce the complexity of the network design, we designed two common structures at different levels. These two structures measure a detection module and another small module, referred to as a picture-generation module, to obtain a feature map of the target. A convolution layer was applied to the input options to scale back the number of channels [29]. It was then processed using a residual structure to extract high-quality options. Finally, by combining completely different levels of feature maps, we built identical scales of feature maps. When a feature map was processed via deconvolution, the channel size of the map was reduced by half.

We created the following design for the division of branches. (1) For branches with an output depth, we allocated a complete full-scale detection branch, including a three-level detection module and an image generation module. (2) For modules that output 2D probabilities and links, we allocated only a shared detection branch, which included a three-level detection module and image-generation module. We tested a variety of network structures and chose the scheme described above as it balances performance and accuracy. We attempted to merge the 3D and 2D branches by providing only three different image-generation modules to output three different types of information. The network performance was poor after testing. After analyzing the network output, we believe that because of the difficulty of 3D branching, the network attempted to learn the tasks of 3D branching as much as possible during the optimization process, resulting in the inability

of the shared parts to provide high-quality 2D feature data. A possible solution was to share only the ResNet Backbone at the forefront, which implied that there were two sets of deconvolution parts. After testing, the performance improvement in this scheme was insufficient to offset the expansion of the network parameters [30,31].

*2.2. Network Postprocessing*

The postprocessing process consists of four steps. (1) The detection module, which uses a convolution operation to quickly find the region with the highest local probability in the heat map, which will be a joint point. (2) Point linking, which generates a group of joint pair-supported heat maps and offset maps. Subsequently, the joint pairs are ranked, and the doable joint connections are calculated with the best total chance. (3) Depth adaptation which calculates the relative proportion of the depth of world space. This process converts mismatched coordinate spaces into the same coordinate space. (4) Time filtering, where, depending on the application, if necessary, time filters are used to smooth the 3D pose results (Figure 3).

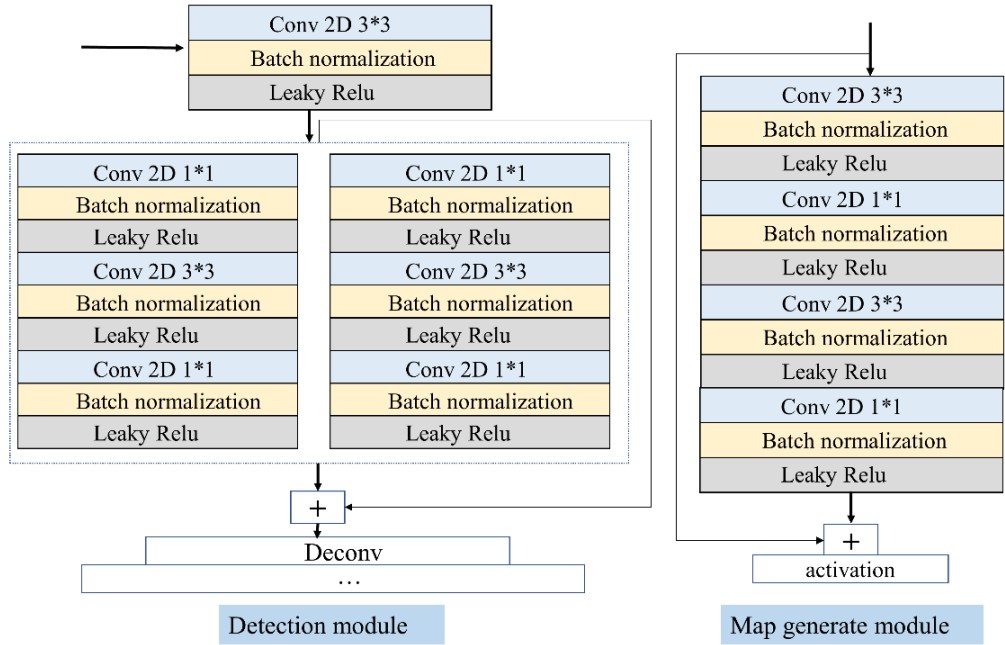

**Figure 3.** Specific structure diagram of detection module and image-generation module. "*" is a multiplication sign.

To connect the joint points, we first calculate the position of the target joint point.

$$(TargetX_j, TargetY_j) = (X_j + OffsetX_j) \tag{9}$$

Next, we searched for a circular region centered on $(TargetX_j, TargetY_j)$, with $R_{search}$ as the radius. The specific calculation method was as follows:

$$l = \sqrt{OffsetX_j{}^2, OffsetY_j{}^2}$$
$$R_{search} = \max(l \times a + (1 - a), 1) \tag{10}$$

For practical applications, we set $a = 0.3$. After testing, this value was found to be highly accurate.

We wanted to scale the depth with the values $scale_{depth}$, so that the coordinates were expressed in the same units.

$$depth = \sqrt{(P_x - C_x)^2 + (Py - Cy)^2 + (P_z - C_z)^2} \tag{11}$$

$$\begin{bmatrix} u_x \\ u_y \\ 1 \end{bmatrix} = \begin{bmatrix} f_x & 0 & C_x \\ 0 & f_y & C_y \\ 0 & 0 & 1 \end{bmatrix} \begin{bmatrix} r_{11} & r_{12} & r_{13} & t_1 \\ r_{21} & r_{22} & r_{23} & t_2 \\ r_{31} & r_{32} & r_{33} & t_3 \end{bmatrix} \tag{12}$$

However, we found a probabilistic correlation between the size and depth of the individuals. We assume that the size of the person is normal. Linear regression and visualization demonstrated a strong correlation between human 2D size (width and height) and $\text{scale}_{\text{depth}}$. Therefore, we use fast calculations to obtain the $\text{scale}_{\text{depth}}$ and multiply it by the depth. This allowed us to estimate the correct position of the human joints in the camera space without obtaining camera information.

## 3. Training Experiment

The JTA dataset was used for training in this study [29,30]. This dataset includes a large number of characters with varied postures and costumes for use, as well as a large number of scenes that provide various weather conditions. Therefore, by combining this information, we can effectively generate indoor and outdoor 3D pose recognition datasets that satisfy our requirements.

We initially trained two branches of the second singly mistreated Microsoft COCO 2014 dataset [27]. This is often similar to coaching a second recognition network. Because only one branch participates in coaching, the number of parameters decreases; therefore, the range of layers of the network is additionally reduced, leading to an associated degree of improvement within the speed of forward and backward propagation of the network. Usually, we train 50–100 epochs for the initialization of the network [2,3]. We used the JTA and Microsoft COCO 2014 datasets for joint coaching (Figure 4).

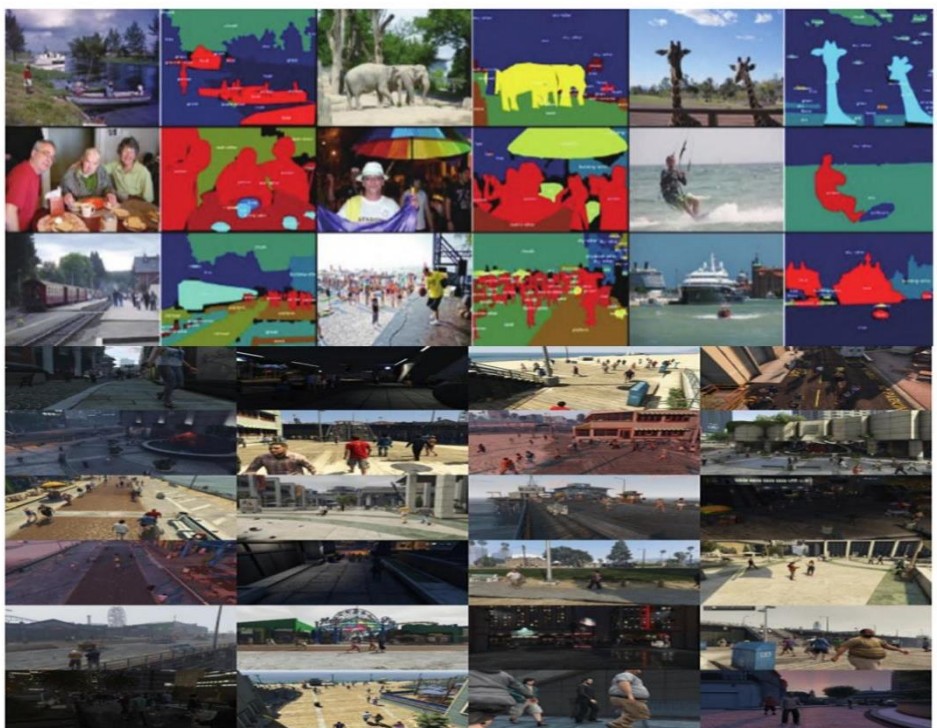

**Figure 4.** JTA and Microsoft COCO 2014 dataset case image.

The entire training system was divided into four modules: data, network, training, and visualization. Various modules are weakly coupled; for example, visualization modules can be used for the separate debugging of data modules. The main acceleration design is located in the data module. For the data module, we first designed multiple dataset interfaces. Our training dataset had two formats; therefore, it was necessary to convert them

into the same format using the dataset interface. Subsequently, the Tensor Pack [19,31] parallel sample generator was used to load the data from these dataset interfaces and integrate them into the final sample (Figure 5). Typically, in neural network training tasks, it is necessary to process large amounts of data. Data such as images and videos usually occupy a large amount of space, and it is not practical to load and complete data processing in one step. The use of a Tensor Pack yields an on-demand effect [32].

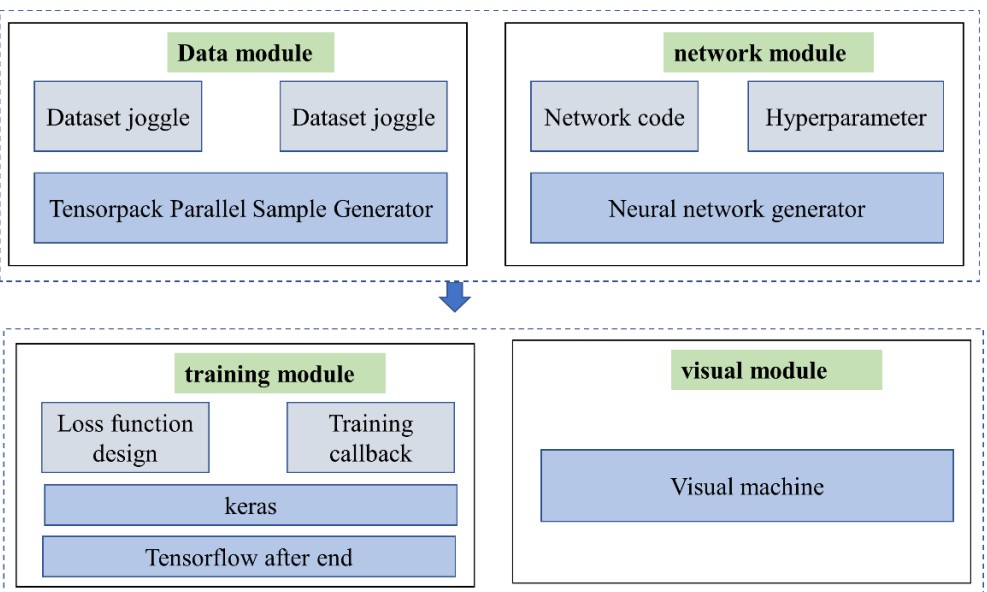

**Figure 5.** Architecture of the training process.

First, we extracted the metadata of the entire dataset. For example, for each sample, data such as joint point positions were extracted, converted into database files, serialized, and placed on a hard drive. This portion of the metadata occupies less space (approximately less than 1 GB). Therefore, it can be directly loaded into the memory to increase the reading speed of the metadata. This part of the metadata was placed in the shared memory for read-only access through multiple processes. The image files in our dataset occupy a large space; therefore, we placed them on a hard drive [33–35].

Subsequently, based on the metadata, region selection was performed on the larger image being read and multiple small regions were sampled from it. Finally, we sent the organized batch data to TensorFlow and uploaded them to the GPU for current batch training. Owing to the adoption of multiple processes in our solution, except for the main training process, other processes still performed data augmentation and preparation, eliminating the need for waiting for the GPU. We observed that by the end of the current batch training, the data required for the next batch were generally prepared, thereby significantly reducing the data preparation time. Numba is a JIT compiler hosted in Python as a package [36,37]. With LLVM, Numba can convert specific Python languages and Numpy calculations into a high-performance machine code. It can generate a highly parallelized Python code that supports both the GPU and CPU simultaneously. This solution can generate machine instructions that bypass the Python global interpreter locks or directly accelerate specific loops. After undergoing Numba JIT conversion, our code retains only the interface part of Python. The current Python interface is forwarded to the JIT-compiled parallelization code for calculation and the result is returned. This part of the parallelized code is not limited by the Python interpreter, and can be parallelized as much as possible. With the optimization mentioned above, our final training system achieved an acceleration of 4x. This enabled us to iterate the network design quickly [38,39].

## 4. Results and Discussion

### 4.1. Error Analysis of Model

We evaluated our approach using the JTA dataset. We used the mean value of the position error per joint (MPJPE) to determine the performance of our network. MPJPE is the mean Euclidean distance between the ground truth and the prediction for multiple joints [40]. The lower the value, the better the performance. Table 2 presents the results of the study. Because we tend to test image-based performance, we do not use a time–electric sander. Because little thought has been given to speed-based 3D creation solutions, we compared them with alternative solutions. Alternative RGB image-based solutions do not target speed and cannot succeed in period performance. Some strategies also use 2D poses. This implies that the standard and speed of the preprocessing methodology can significantly affect the performance of the subsequent 3D estimation strategies. We tested these strategies by adding a typical 2D perspective estimation system, OpenPose, for live speed and accuracy. Open pose requires all interconnected limbs to generate pose recognition results directly for all individuals. Open pose generates two parts of output: one part calibrates the type of area in the input screen, including limb types such as elbows and necks, as well as the background; while the other part of it encodes "connectivity" information [41,42].

**Table 2.** Performance comparison between our methods and previous ones. '-' indicates that this solution is not designed for real-time, so demonstrating its speed is meaningless, or its speed is much lower than real-time.

| Methods | Input | MPJPE (mm) | Velocity |
|---|---|---|---|
| Zhou et al. [34] | Video | 65.1 | - |
| Martinez et al. [35] | 2D pose | 64.1 | - |
| Open Pose (high accuracy mode)/Martinez et al. [35] | RGB image | 71.2 | 834.0 |
| Open Pose (high speed mode)/Martinez et al. [35] | RGB image | 83.1 | 114.2 |
| LCR-Net [36] | RGB pose | 83.1 | - |
| This study | RGB pose | 113.2 | 47.1 |

However, many situations in the test dataset were unsuitable for our design goals. The test dataset contained many people of different sizes, with some human bodies obstructing each other or being obscured by the foreground and background. This is significantly different from the application scenarios used in our design. Therefore, we also used test sets that were more similar to games or entertainment to test our work. The resulting image is shown in Figure 6.

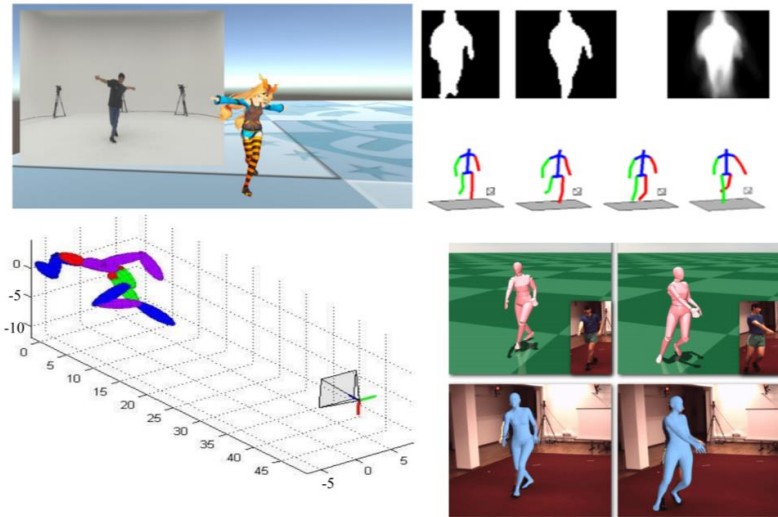

**Figure 6.** 3D pose recognition.

### 4.2. Performance between Multiple Datasets

The accuracy of the 17 human keypoints predicted by the five models on Microsoft COCO 2014 is shown in Table 3, while Table 4 lists the accuracy of the 15 human keypoints predicted by them on the CMU Panoptic test set (as the front image of the ankle joint in the CMU Panoptic dataset is not visible, the final prediction result has 15 keypoints). The last columns of Tables 3 and 4 show the average accuracy of the predictions of the keypoints. It can be observed that the prediction accuracy of the basic model's keypoints is significantly lower than that of the other four models. The introduction of an anchor-pose-based regression module for the design and use of anchor frames has long been considered an essential part of high-precision object detection (Faster RCNN/SSD/YOLOv2&v3). The anchor box represents the initial state of the candidate target, and is an effective method for obtaining the potential distribution area of the target [37,38]. By combining a feature pyramid network (FPNet) and anchor-box-based detectors (FPN/RetinaNet/RefineDet), the accuracy of the method was improved to new heights [43,44], which in turn improved the prediction accuracy of each keypoint. The accuracy of the model directly projected from a 3D pose to a frontal 2D pose was slightly higher than that of adding an anchor pose. The introduction of a module that integrates 3D pose data inputs has improved the basic model, especially at keypoints such as the left ear, right ear, left shoulder, left crotch, and right crotch. The average accuracy of predicting key points is the highest when introducing a regression module based on the anchor pose and a module fused with 3D pose data input, with only a few keypoints showing little decrease. Overall, the FPNet has the highest prediction accuracy, but there is still room for improvement [45].

**Table 3.** Accuracy of different models on Microsoft COCO 2014 (%).

| Joint Point | Base | 2D-3D Pose | Base + Anchor Pose | Base + Anchor 3D Pose | Our Method |
|---|---|---|---|---|---|
| Nose | 32.9 | 44.0 | 43.2 | 49.3 | 55.9 |
| Left eye | 38.6 | 46.7 | 45.6 | 50.9 | 58.6 |
| Right eye | 34.3 | 46.7 | 40.9 | 52.3 | 65.1 |
| Lest ear | 53.2 | 62.5 | 61.0 | 66.8 | 61.0 |
| Right ear | 49.9 | 63.9 | 60.9 | 68.9 | 65.5 |
| Left shoulder | 44.8 | 73.5 | 14.3 | 73.2 | 75.3 |
| Right shoulder | 45.0 | 73.5 | 18.6 | 33.0 | 74.5 |
| Left elbow | 7.2 | 32.1 | 5.5 | 28.4 | 35.5 |
| Right elbow | 10.4 | 28.4 | 5.0 | 7.0 | 33.5 |
| Left wrist | 3.4 | 6.9 | 42.1 | 5.4 | 6.5 |
| Right wrist | 2.8 | 5.3 | 33.6 | 54.9 | 5.5 |
| Left hip | 41.5 | 48.9 | 19.5 | 49.1 | 57.4 |
| Right hip | 19.2 | 45.1 | 18.5 | 25.8 | 50.3 |
| Left knee | 17.5 | 23.6 | 19.5 | 26.4 | 26.4 |
| Right knee | 11.8 | 25.9 | 18.6 | 10.6 | 27.3 |
| Left ankle | 7.8 | 11.9 | 16.6 | 14.0 | 8.1 |
| Right ankle | 3.7 | 13.6 | 7.9 | 11.2 | 12.2 |
| Average | 25.0 | 39.1 | 33.1 | 41.2 | 42.1 |

**Table 4.** Accuracy of different models on the CMU Panoptic dataset (%).

| Joint Point | Base | 2D-3D Pose | Base + Anchor Pose | Base + Anchor 3D Pose | Our Method |
|---|---|---|---|---|---|
| Nose | 22.2 | 55 | 60.1 | 61.7 | 67.2 |
| Left eye | 20.1 | 56.9 | 58.2 | 58.9 | 70.5 |
| Right eye | 22.5 | 49.7 | 54.8 | 56.8 | 64.0 |
| Lest ear | 52.9 | 62.5 | 55.5 | 56.8 | 70.5 |
| Right ear | 35.1 | 58.9 | 56.5 | 55.5 | 56.1 |
| Left shoulder | 51.5 | 63.5 | 58.2 | 56.2 | 63.3 |

**Table 4.** *Cont.*

| Joint Point | Base | 2D-3D Pose | Base + Anchor Pose | Base + Anchor 3D Pose | Our Method |
|---|---|---|---|---|---|
| Right shoulder | 12.6 | 63.5 | 58.2 | 57.5 | 23.1 |
| Left elbow | 11.8 | 22.1 | 17.1 | 57.5 | 22.5 |
| Right elbow | 7.6 | 20.5 | 18.5 | 17.8 | 12.2 |
| Left wrist | 8.1 | 9.0 | 8.9 | 18.5 | 14.1 |
| Right wrist | 11.9 | 9.8 | 8.9 | 8.9 | 50.5 |
| Left hip | 7.6 | 49.0 | 41.8 | 8.9 | 41.9 |
| Right hip | 7.5 | 44.6 | 46.6 | 44.5 | 39.9 |
| Left knee | 22.6 | 33.5 | 40.4 | 41.8 | 31.2 |
| Right knee | 26.5 | 49.6 | 38.3 | 39.1 | 45.9 |
| Left ankle | 22.1 | 10.2 | 38.6 | 41.7 | 46.1 |
| Right ankle | 19.2 | 6.3 | 42.3 | 41.6 | 46 |
| Average | 25.6 | 39.0 | 40.1 | 41.7 | 46.2 |

Some typical qualitative examples of application of models on the Microsoft COCO 2014 dataset, as shown in Figure 7, exhibit significant attitude changes and scale variances. The first row represents the input human body image, the second row represents the annotated positive posture in red, and the blue row represents the output of the model. It can be observed that with the introduction of the anchor pose regression module and 3D pose fusion module, the extracted frontal posture also had good expression for some backside human body images or side human body images with a large angle with the front side human body image. In addition, it performed well for images with missing data caused by human self-occlusion [46,47].

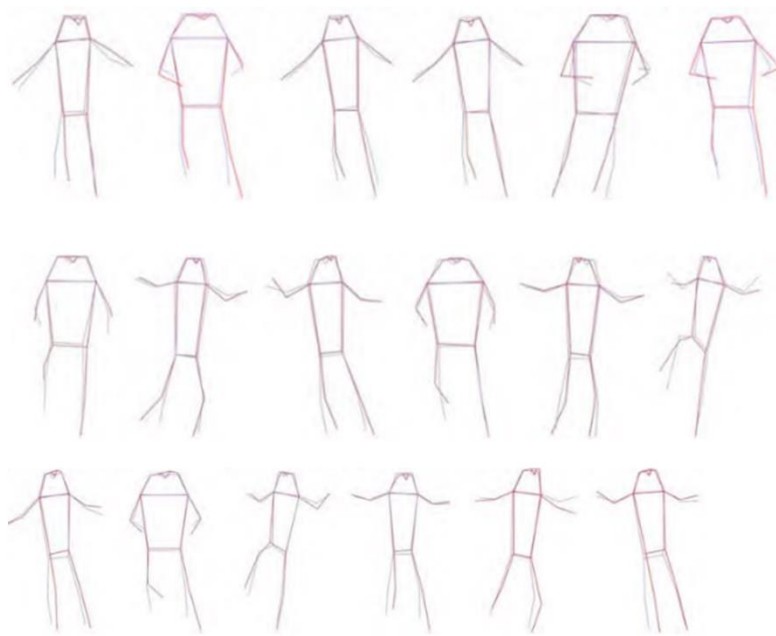

**Figure 7.** Positive attitude estimation for multi-views based on the Microsoft COCO 2014 dataset.

However, there is still room for improvement in the prediction of some keypoints in the human body, particularly the wrists, ankles, and knees. After analysis, it is believed that the reason for this is that the model performance has not yet achieved accurate predictions for all joints; however, there is severe occlusion of some joints in the input images, which affects the prediction accuracy. Partial experimental results for the CMU Panoptic dataset are shown in Figure 8. The prediction results obtained using this dataset were more accurate. Among them, there are fewer action occlusions, the accuracy of the prediction results is relatively high, and the prediction of some joint points is not very accurate. However, the

overall prediction trend is accurate. In summary, the FPNet can effectively predict human frontal posture, but the performance still has room for improvement.

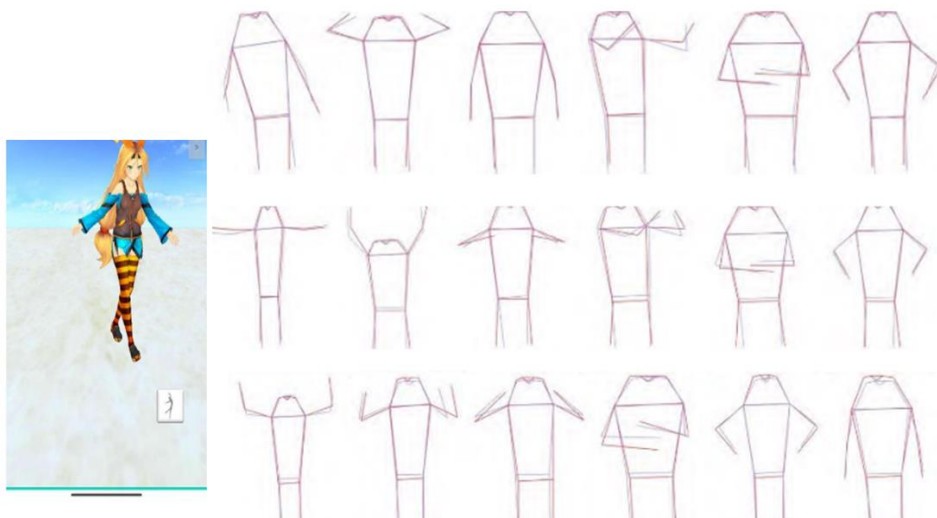

**Figure 8.** Back pose estimation on the CMU Panoptic dataset for multi-views.

*4.3. Comparative Experiment Analysis*

This section evaluates the proposed method using the Human3.6M motion capture dataset. Similar to VideoPose, this section also uses the MPJPE indicator for evaluation. For Human3.6M, the network structure parameters are as follows: The number of input joint points N is 17. The length, T, of the 2D joint-point sequence is 243. The main network consists of four convolutional blocks, the first two being convolutional blocks A and the last two being convolutional blocks B. The convolutional kernel size and hole convolution scale are the same as those in the examples in Figures 3–6. In the local convolution, the number of output channels for each group of convolutions was 64. In global convolution, the number of output channels was 1088. The training parameters were as follows: the batch size was 256, the ranger was used as the optimizer, the initial learning rate was 0.001, and the learning rate of each iteration decayed by 95%. Eighty iterations were performed.

In the case of using 2D annotations as the network input, this chapter compares it with previous studies on MPJPE metrics, the results of which are shown in Figure 9. The algorithm proposed in this chapter outperforms previous algorithms in predicting all actions, especially when compared with the baseline VideoPose method in this framework. The results showed a significant improvement, with an average error reduction of 4.8 mm. The experimental results show that the method proposed in this section has significant advantages and significantly improves the accuracy of 3D human pose estimation.

To verify the robustness of this framework, we used the cascaded pyramid network (CPN) prediction results as inputs to the network. Chen et al. [44] proposed CPN to facilitate the accurate location of occluded and invisible points, utilizing multi-level features to clearly solve the detection problem of "difficult points." Compared with pure 2D annotation, the predicted results of the CPN are not accurate and contain certain noise, which impacts the subsequent 3D human pose estimation. However, in a horizontal comparison, the algorithm proposed in this section still has significant advantages over the previous methods [48]. With the MPJPE index, the average error was reduced to 44.8 mm, whereas with the P-MPJPE index, it was reduced to 35.2 mm. Compared to the baseline method, the two indicators decreased by 2.0 mm (4.4%) and 2.6 mm (6.9%), respectively (Figure 10).

The P-MPJPE index shows a greater decrease than the MPJPE index, which is nearly twice as large, indicating that this framework has a stronger regression ability in terms of action accuracy without considering scaling and rotation (Figure 11).

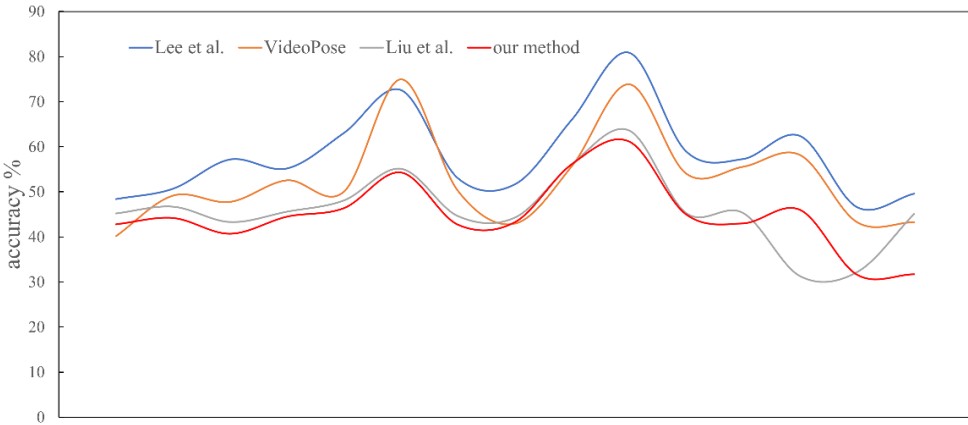

**Figure 9.** Comparison of MPJPE metrics (mm) for 15 actions on the Human3.6M dataset (2D annotations as input), where the methods refer to Lee et al. [37], Video Pose [37,38], and Liu et al. [39], respectively. The description of the Y axis is accuracy rate %.

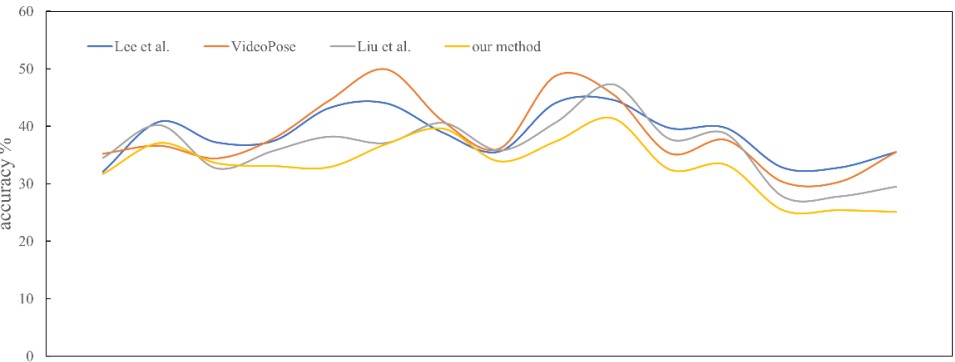

**Figure 10.** Comparison of MPJPE metrics (mm) for 15 actions on the Human3.6M dataset (CPN prediction results as input), where methods refer to Lee et al. [37], VideoPose [38], and Liu et al. [39], respectively. The description of the Y axis is accuracy rate %.

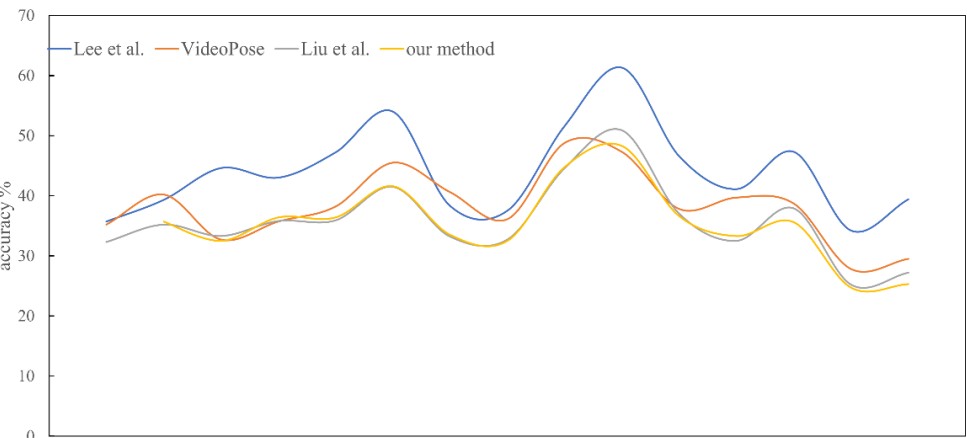

**Figure 11.** Comparison of P-MPJPE indicators (mm) for 15 actions on the Human3.6M dataset (CPN prediction results as input), where methods refer to Lee et al. [37], VideoPose [38], and Liu et al. [39], respectively. The description of the Y axis is accuracy rate %.

## 5. Conclusions

This study proposes a system for 3D pose estimation. It includes a multitasking and multilevel detection neural network that can directly input and output 2D joint positions, 2D joint links, and 3D joint depth information from the images. Postprocessing was used to construct 3D poses for each animation. Our RGB-image-based solution reduces accuracy by only 33 mm and can achieve a speed of 21 fps on an RTX 2080. Our system demonstrates potential for real-time 3D pose estimation of multiple animations using regular RGB cameras. In summary, the 3D-pose-estimation scheme implemented in this study can capture the 3D pose of a human body using ordinary RGB cameras.

The model proposed in this study demonstrates more accurate recognition ability and good execution performance based on classic character pose recognition. The improved algorithm improves the recognition accuracy by approximately 3.5% and the performance by 8–10 times, which is significantly superior to other classic algorithms. Experimental data demonstrate the advantages of this algorithm in recognizing human body movements and postures.

The multilevel detection theory used in this study was based on the size of the object occupied in the input image. This is understandable under normal conditions; however, in situations where a part of the character is outside the screen, the detection level may be incorrect. At the same time, the basis of the multilevel detection theory is that multiple levels share tasks and cooperate with one another, which has also been verified in network visualization research. However, the balance of tasks at multiple levels is currently a weak link in multilevel detection theory. Analyzing the data distribution of the actual target tasks and balancing the detection tasks at each level is a solution that can be considered in the future.

**Author Contributions:** Conceptualization, supervision, methodology, project administration, investigation, resources, writing—original draft preparation, W.D. and W.L.; data curation, validation, writing—review and editing, W.D. and W.L. All authors have read and agreed to the published version of the manuscript.

**Funding:** This research received no external funding.

**Institutional Review Board Statement:** Not applicable for studies not involving humans or animals.

**Informed Consent Statement:** Not applicable.

**Data Availability Statement:** The data used to support the findings of this study are available from the corresponding author upon request.

**Conflicts of Interest:** We declare that we have no financial and personal relationships with other people or organizations that can inappropriately influence our work. There is no professional or other personal interest of any nature or kind in any product, service and/or company that could be construed as influencing the position presented in, or the review of, the manuscript entitled.

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
