# Peer review of "High Speed and Accuracy of Animation 3D Pose Recognition Based on an Improved Deep Convolution Neural Network"

_applsci, doi:10.3390/app13137566_

Round 1
Reviewer 1 Report
The construction of this scientific work and research is well conceived, but not adequately and comprehensively described. Abstract already lacks a clear description of whether the focus is on facial or body pose recognition or both, because in the following text only recognition of body poses, i.e. mostly limbs, is mentioned.
Datasets MSCOCO 2014, CMU Panoptic, 18, Human3.6M that are indicated in the Abstract are not described in detail or informatively in the Introduction chapter. Also, for the indicated datasets, traceability and reference information about who first started using them was not presented. This is essential for the understanding of the research as well as the interest and traceability of the readers.
In general, the text contains a multiple tipfellers, underdeveloped words and indexing errors in table matrices, equations. Let's go in order:
1. In equation 1, the entry bracket is missing from the notation;
2. In equation 2, the capital letter Y sign is missing from the record;
3. In equation 5, next to the sign of the capital letter Z in the record number 1 is the index (Subscript);
4. In equation 11, after the first parentheses number 2 is a superscript;
5. Line 26,160, 167, 169, 230, 235, 231, ect. there is a small letter at the beginning of the sentence.
6. Line 172 description and reference for the term ResNet34 is missing;
7. Line 225 description and reference for the term JTA dataset standard is missing;
8. Line 230 description and reference for the term MS coconut 2014 is missing dataset standard;
9. From line 237-243, the section of the text should be paraphrased because it is a list;
10. Line 252 the description and reference for the phrase "Tensor Pack parallel sample generator" is missing;
11. Line 262In there is an unclear sentence and it is necessary to paraphrase and clarification;
12. Line 271 is missing the description and reference for the phrase "Numba JIT just-in-time Conversion".
13. Line 280 - the description and reference of earlier use for the term and standard of MPJPE are missing;
14. Line 289 - the description and reference of the earlier use of the term and the OpenPose Python function are missing;
15. Line 303 & 307 - the period at the end of the sentence is missing;
16. Line 304 - the description and reference of the earlier use of the term and the BJUT Taichi test are missing;
17. Line 305 - the description and reference of the earlier use of the term "CMU Panoptic test" are missing;
18. Line 310 - the description and reference of the earlier use of the term "APB & Anchor pose-based regression module" are missing;
19. Line 347 - the description and reference of the earlier use of the term "Human3.6M motion capture dataset" are missing;
20. Line 368 & 380 - missing space after comma in a sentence;
21. Line 370 - the description and reference of the earlier use of the term "CPN prediction" are missing;
22. Figure 9, 10 and 11 - the description of the Y axis is: "accuracy" or "error rate" according to the improved procedure of the author's method;
23. In general, the entire paper should be written in the third person singular or plural for easier understanding and exclusion of subjectivity (... it was designed, it was made, it was calculated, ect.).
Clarifying the meaning of several used terms of data set standards, Python functions and referencing them will significantly increase the list of reference literature that is necessary for monitoring and understanding as well as verifying the veracity of this approach and research.
Finally, it is necessary to improve the written language in accordance with the writing style in English.
For the understanding, acceptance and popularization of scientific research, a description of the scientific research supported by precise and exact descriptions of the used methods, procedures, standards, ready-off functions, datasets and practical experiences is crucial.
Dear Editor-in-Chef,
After a complete and critical analysis of the topic and procedures used in the subject paper, I wrote a report for the authors. I advise that the paper be written in the third person in order to avoid personal subjectivity in the expression of the author and to avoid the personification of scientific research.
It is necessary to expand the list of reference literature with recently published Open Access works in the field.
It is also necessary to significantly paraphrase the text in accordance with the English style of writing and create a high-quality translation of the text into English.
I advise the publication of this work after significant corrections, additions and correct translations of the text.
PhD V. Tudić, prof.
Deputy President of the Karlovac County Innovation Association
Author Response
We are so grateful for reviewing our manuscript from the two reviewers. The suggestions are very helpful and significant. We substantially improve our manuscript according to these comments, which are marked as red in the revised version. Language of this paper is thoroughly improved including style. Details point-by point is given as follows. Where reviewer’s comments are marked as black, the answer is marked blue.
Reviewer#1
- The construction of this scientific work and research is well conceived, but not adequately and comprehensively described. Abstract already lacks a clear description of whether the focus is on facial or body pose recognition or both, because in the following text only recognition of body poses, i.e. mostly limbs, is mentioned.
Response and done:
In fact, this study referred both of the body pose and facial recognition of animation, including limbs, eyes, noses, etc. we have improved the abstract as following:
“Pose recognition of character animation is an important research content in the field of computer graphics. However, currently using traditional artificial intelligence algorithms to recognize animation gesture faces problems of low accuracy and speed. Therefore, in order to overcome the above problems, this paper proposes a real-time 3D pose (facial and body pose) recognition system based on deep convolutional neural networks, and further designs a single purpose 3D pose estimation system……”
Datasets MSCOCO 2014, CMU Panoptic, 18, Human3.6M that are indicated in the Abstract are not described in detail or informatively in the Introduction chapter. Also, for the indicated datasets, traceability and reference information about who first started using them was not presented. This is essential for the understanding of the research as well as the interest and traceability of the readers.
Response and done:
We have added a sentence described the four posture datasets used in the Introduction of this study, we also added the corresponding references which is source of the dataset. Details see followings:
Posture dataset can enable to reality of detection of targets, posture, semantic seg-mentation, strength segmentation, etc. Recently, a large number of posture datasets have been widely used to experiment comparison. The Microsoft COCO 2014 dataset was con-structed by Lin et al. (2014) [27], which is a set of 2D image datasets provided by Microsoft, widely used for tasks such as object detection, pose detection, semantic segmentation, and strength segmentation. It contains key point information of the human body in various scenarios. For the COCO pose detection dataset, there are a total of 120000 sample images, including approximately 64000 sample images of the human body. For these images, the positions of each key point of the human body (2D coordinates) and visible and invisible (represented by 0, 1, and 2 respectively for unlabeled, invisible, and visible situations) were manually annotated. The CMU Panoptic dataset is a typical case study [28]. The Joo et al. (2015) [28] constructed a spherical dome area where 480 VGA cameras were placed, each with a resolution of 640 x480, and synchronized using a hardware clock. The captured frame rate is 25 fps. At the same time, 31 high-definition cameras were installed to record images at a resolution of 1920 x 1080, operating at 30 fps while ensuring time synchronization with VGA cameras. In order to capture the 3D pose of the human body, 10 Kinect v2 cameras were used to complete the 3D pose capture. The Kinect V2 camera has resolutions of 1920 x 1080 (RGB) and 512 x 424 (depth). The KinectV2 camera operates at 30fps and maintains the same time as the previous two cameras. The source of JTA data collection is Rockstar's game Grand Theft Auto 5 [29]. It adopts a physics-based rendering architecture that can produce high-precision character animations. And it contains a large number of characters with different postures and costumes for use, as well as a large number of scenes and provides various weather conditions. Human3.6M [30] is currently recognized as the largest 3D human posture dataset, which includes 15 daily life actions such as walking, eating, sitting, and walking the dog. Its actions were performed by 11 professional actors wearing motion capture equipment in the laboratory environment, and were recorded by four synchronous cameras at 50Hz, with a total of about 3.6 million video pictures. The data of 7 actors includes 3D annotations.
In general, the text contains a multiple tipfellers, underdeveloped words and indexing errors in table matrices, equations. Let's go in order:
- In equation 1, the entry bracket is missing from the notation;
We have improved it:
|
(1) |
(1) |
- In equation 2, the capital letter Y sign is missing from the record;
We have improved it:
|
(2) |
(2) |
- In equation 5, next to the sign of the capital letter Z in the record number 1 is the index (Subscript);
We have improved it:
|
(5) |
(5) |
- In equation 11, after the first parentheses number 2 is a superscript;
We have improved it:
(11)
- Line 26,160, 167, 169, 230, 235, 231, ect. there is a small letter at the beginning of the sentence.
We have checked them and improved them.
- Line 172 description and reference for the term ResNet34 is missing;
We added a sentence described the ResNet:
In the 2015 ILSVRC competition, Resnet et al. proposed Residual Network [31], which achieved good results. Resnet further refined image feature maps with fewer parameter quantities and deeper network structure, resulting in image feature representations with better information representation capabilities. We use ResNet34 because the face, however compress the channel in 0.5. The elaborated structure is shown within the Table 1.
- Line 225 description and reference for the term JTA dataset standard is missing;
The description of JTA is introduced in the Introduction section.
- Line 230 description and reference for the term MS coconut 2014 is missing dataset standard;
We have described the details (source and functions, etc.) of Microsoft COCO 2014 dataset in the Introduction section.
- From line 237-243, the section of the text should be paraphrased because it is a list;
We deleted this paragraph for similarity.
- Line 252 the description and reference for the phrase "Tensor Pack parallel sample generator" is missing;
We have added the corresponding description about Tensor Pack as following:
Subsequently, the Tensor Pack [18] parallel sample generator will load the data from these dataset interfaces and integrate them into the final sample (Figure 5). Usually, in neural network training tasks, it is often necessary to process a large amount of data. Data such as images and videos usually occupy a large amount of space, and it is not practical to load and complete all data processing in one go. Using the Tensor Pack can achieve an on-demand effect.
- Line 262In there is an unclear sentence and it is necessary to paraphrase and clarification;
We improved the paragraph to following:
We first extract the metadata of the entire dataset. For example, for each sample, ex-tract data such as joint point positions, convert them into database files, serialize them, and place them on the hard drive. This portion of metadata takes up less space, approximately within 1GB. Therefore, it can be directly loaded into memory to speed up the reading speed of metadata. This part of metadata is placed in shared memory for read-only access by multiple processes. And the image files in our dataset occupy a large space, so we place them on the hard drive.
- Line 271 is missing the description and reference for the phrase "Numba JIT just-in-time Conversion".
We added a sentence described the Numba JIT as followings:
Numba [39,40] is a JIT compiler hosted in the Python language as a Python package. With LLVM, Numba is able to convert specific Python languages and Numpy calculations into high-performance machine code. Can generate highly parallelized Python code that sup-ports both GPU and CPU simultaneously. This solution can generate machine instructions that bypass Python global interpreter locks, or directly accelerate specific for loops. After undergoing Numba JIT conversion, our code will only retain the interface part of Python. Calling the current Python interface will be forwarded to the JIT compiled parallelization code for calculation, and then the result will be returned. This part of the parallelized code is not limited by the Python interpreter and can be parallelized as much as possible. With the optimization mentioned above, our final training system achieved 4x acceleration. This enables us to quickly iterate on network design [39].
- Line 280 - the description and reference of earlier use for the term and standard of MPJPE are missing;
We added a corresponding description:
MPJPE is mean Euclidean distance between ground truth and prediction for multiple joints [41], the lower value is, the better performance is.
- Line 289 - the description and reference of the earlier use of the term and the OpenPose Python function are missing;
We added the corresponding description:
Open pose is based on obtaining all interconnected limbs to directly generate pose recog-nition results for all individuals. Open Pose will generate two parts of output: one part calibrates the type of each area in the input screen, including limb types such as elbows and necks, as well as the background; Part of it encodes "connectivity" information [42].
- Line 303 & 307 - the period at the end of the sentence is missing;
We have improved them.
- Line 304 - the description and reference of the earlier use of the term and the BJUT Taichi test are missing;
We have replaced it into Microsoft COCO 2014 data set, and results are also improved.
- Line 305 - the description and reference of the earlier use of the term "CMU Panoptic test" are missing;
We have described the dataset in the Introduction.
- Line 310 - the description and reference of the earlier use of the term "APB & Anchor pose-based regression module" are missing;
We have added the corresponding description:
“…for a long time, the design and use of anchor frames have been considered an essential part of high-precision object detection (Faster RCNN/SSD/YOLOv2&v3). The anchor box represents the initial state of the candidate target and is an effective way to obtain the po-tential distribution area of the target. By combining FPN (Feature Pyramid Network) and anchor box based detectors (FPN/RetinaNet/RefineDet), the accuracy of the method has been pushed to new heights [43],…”
- Line 347 - the description and reference of the earlier use of the term "Human3.6M motion capture dataset" are missing;
We have described it in the Introduction section.
“…Human3.6M [30] is currently recognized as the largest 3D human posture dataset, which includes 15 daily life actions such as walking, eating, sitting, and walking the dog. Its ac-tions were performed by 11 professional actors wearing motion capture equipment in the laboratory environment, and were recorded by four synchronous cameras at 50Hz, with a total of about 3.6 million video pictures. The data of 7 actors includes 3D annotations…”
- Line 368 & 380 - missing space after comma in a sentence;
We have improved it.
- Line 370 - the description and reference of the earlier use of the term "CPN prediction" are missing;
We have added the corresponding description as following:
“…To verify the robustness of this framework, we use CPN prediction results as inputs to the network. Chen et al. [44] proposed a Cascaded Pyramid Network (CPN) to solve the problem of difficult to accurately locate occluded and invisible points, utilizing multi-level features to clearly solve the detection problem of "difficult points…”
- Figure 9, 10 and 11 - the description of the Y axis is: "accuracy" or "error rate"according to the improved procedure of the author's method;
We have improved them.
- In general, the entire paper should be written in the third person singular or plural for easier understanding and exclusion of subjectivity (... it was designed, it was made, it was calculated, ect.).
Response and done:
Thank you for your careful suggestions, which are indeed helpful for our study is noticed by readers.
We have thoroughly revised them in the revised version where you can find them marked as red.
Clarifying the meaning of several used terms of data set standards, Python functions and referencing them will significantly increase the list of reference literature that is necessary for monitoring and understanding as well as verifying the veracity of this approach and research.
We have clearly introduced the four datasets that used in our study in the Introduction section, including source and wide usage of datasets which are indicated by reference literature. The main methods and python functions used in this study have also introduced in the section Methodology and referred them in the literature list.
Finally, it is necessary to improve the written language in accordance with the writing style in English.
Thanks, we have invited the professional language company to help to improve English expressions in present study, if you think they are still needed to be modified, please let us known it.
For the understanding, acceptance and popularization of scientific research, a description of the scientific research supported by precise and exact descriptions of the used methods, procedures, standards, ready-off functions, datasets, and practical experiences is crucial.
Thank you for your suggestions, we have improved them in the revised paper, if there are still having problems, please let us known them.
Comments on the Quality of English Language
Dear Editor-in-Chef,
After a complete and critical analysis of the topic and procedures used in the subject paper, I wrote a report for the authors. I advise that the paper be written in the third person in order to avoid personal subjectivity in the expression of the author and to avoid the personification of scientific research.
It is necessary to expand the list of reference literature with recently published Open Access works in the field.
It is also necessary to significantly paraphrase the text in accordance with the English style of writing and create a high-quality translation of the text into English.
I advise the publication of this work after significant corrections, additions, and correct translations of the text.
PhD V. Tudić, prof.
Deputy President of the Karlovac County Innovation Association
Response and done:
As for what you noticed in terms of suggestions, we have improved them point by point. We have thoroughly modified the expression of the author. We have added many corresponding literatures in the field (up to number of 48 references), most of them are recently published Open Access works. We have improved the English expression.
Reviewer 2 Report
This paper proposes a real-time 3D pose recognition system based on deep 12 convolutional neural networks, and further designs a single purpose 3D pose estimation system. This is an interesting paper; however it must be improved by taking into account the following points:
i. The related work should contain more recent works.
ii. The main contributions of the paper with respect to the existing literature should be specified.
iii. The title of the paper is “An improved deep convolution neural network-based animation 3D pose recognition”; clearly specify what improvement is done on which base method. The novelty of the paper must be discussed more specifically.
iv. Details about JTA and MSCOCO dataset should be given with references.
v. The proposed method is validated using only two dataset. More datasets should be used to validate the proposed method.
vi. Is there any justification for 50-100 epochs?
There are many grammatical and typographical errors that must be removed.
Author Response
Reviewer#2
This paper proposes a real-time 3D pose recognition system based on deep 12 convolutional neural networks, and further designs a single purpose 3D pose estimation system. This is an interesting paper; however it must be improved by taking into account the following points:
- The related work should contain more recent works.
We have added some recent works related the field, total literatures are up to number of 48.
- The main contributions of the paper with respect to the existing literature should be specified.
We have added our main contributions in the Introduction as followings.
The main contributions of present study are including (1) using a multitasking de-sign method, 2D and 3D poses were successfully recognized in the same network, greatly improving the information transmission and acceleration space of the network; (2) de-scribing the acceleration scheme used for training and the network visualization and runtime visualization system used for debugging, so that the process discussed in this article not only has theoretical value, but also has practical significance in engineering; (3) the system described in this paper has both data acquisition and post data acquisition applications, demonstrating the various roles that deep learning technology can play in the field of computer graphics animation.
- The title of the paper is “An improved deep convolution neural network-based animation 3D pose recognition”; clearly specify what improvement is done on which base method. The novelty of the paper must be discussed more
We modified the title of paper to “High speed and accuracy of animation 3D pose recognition based on an improved deep convolution neural network”.
We have clearly discussed novelty of the study as followings:
“Therefore, this paper proposes a series of technical solutions for animation posture recognition, and output based on neural network structures, which have better performance than traditional methods for animation posture output. In order to overcome the above shortcomings in human posture recognition, this paper proposes three improvements in object recognition, 3D posture estimation, and animation generation.1) Using the concept of "convolution" in signals and systems, a convolutional neural network is pro-posed. Convolutional neural networks can achieve efficient processing of large images by allowing their own neurons to output only according to the surrounding cells within the coverage range. A convolutional neural network generally consists of two parts: the first half is composed of multiple convolutional layers, and the second half is composed of fully connected neurons. The parameters that need to be learned in the convolutional layer of a convolutional neural network are "convolutional kernels" one by one, so the amount of parameters that need to be considered is less than that of traditional neural networks, so as to achieve analysis of complex information with fewer parameters. Therefore, convolutional neural networks have become an attractive deep learning neural network structure for computer vision; 2) The scheme of directly using images as input does not rely on 2D posture data, but directly uses image information to complete 3D posture estimation. This scheme can utilize the light and shadow features of the image to assist in estimation, and therefore can provide more accurate results than the previous scheme. However, the over-all work is more difficult and the system complexity is higher; 3) The researcher attempts to generate real-time character animation that conforms to the characteristics of the sur-rounding environment and character behavior based on the collected 3D posture information, using the surrounding environment and character behavior as input. This technology can greatly reduce the workload of animation creators and simplify the complexity of creating high fidelity virtual character animations. The traditional algorithm adopts the form of violent search, which quickly searches the existing 3D posture database and selects the most matching posture data for output. Such a scheme relies on large memory and heavily on CPU performance.
The main contributions of present study are including (1) using a multitasking de-sign method, 2D and 3D poses were successfully recognized in the same network, greatly improving the information transmission and acceleration space of the network; (2) de-scribing the acceleration scheme used for training and the network visualization and runtime visualization system used for debugging, so that the process discussed in this article not only has theoretical value, but also has practical significance in engineering; (3) the system described in this paper has both data acquisition and post data acquisition applications, demonstrating the various roles that deep learning technology can play in the field of computer graphics animation.”
- Details about JTA and MSCOCO dataset should be given with references.
We have added a paragraph described the main four datasets we used in the Introduction.
Posture dataset can enable to reality of detection of targets, posture, semantic segmen-tation, strength segmentation, etc. Recently, a large number of posture datasets have been widely used to experiment comparison. The Microsoft COCO 2014 dataset was construct-ed by Lin et al. (2014) [27], which is a set of 2D image datasets provided by Microsoft, widely used for tasks such as object detection, pose detection, semantic segmentation, and strength segmentation. It contains key point information of the human body in various scenarios. For the COCO pose detection dataset, there are a total of 120000 sample images, including approximately 64000 sample images of the human body. For these images, the positions of each key point of the human body (2D coordinates) and visible and invisible (represented by 0, 1, and 2 respectively for unlabeled, invisible, and visible situations) were manually annotated. The CMU Panoptic dataset is a typical case study [28]. The Joo et al. (2015) [28] constructed a spherical dome area where 480 VGA cameras were placed, each with a resolution of 640 x480, and synchronized using a hardware clock. The cap-tured frame rate is 25 fps. At the same time, 31 high-definition cameras were installed to record images at a resolution of 1920 x 1080, operating at 30 fps while ensuring time syn-chronization with VGA cameras. In order to capture the 3D pose of the human body, 10 Kinect v2 cameras were used to complete the 3D pose capture. The Kinect V2 camera has resolutions of 1920 x 1080 (RGB) and 512 x 424 (depth). The KinectV2 camera operates at 30fps and maintains the same time as the previous two cameras. The source of JTA data collection is Rockstar's game Grand Theft Auto 5 [29]. It adopts a physics based rendering architecture that can produce high-precision character animations. And it contains a large number of characters with different postures and costumes for use, as well as a large number of scenes and provides various weather conditions. Human3.6M [30] is currently recognized as the largest 3D human posture dataset, which includes 15 daily life actions such as walking, eating, sitting, and walking the dog. Its actions were performed by 11 professional actors wearing motion capture equipment in the laboratory environment, and were recorded by four synchronous cameras at 50Hz, with a total of about 3.6 million video pictures. The data of 7 actors includes 3D annotations.
- The proposed method is validated using only two dataset. More datasets should be used to validate the proposed method.
In fact, we used four datasets validate our method (MSCOCO 2014, CMU Panoptic, Human3.6M and JTA), which are described in the Introduction. Meanwhile we also compared our method with others (Lee et al. [35],Video Pose [36-37], and Liu et al.[38]. References refers to original paper), details can refer to “4.3. Comparative experiment analysis”.
- Is there any justification for 50-100 epochs?
We have added some previous researchers to validate this criterion.
[2]. Zhang Hong,Li Yang,Yang Hanqing,He Bin & Zhang Yu.(2022).Isomorphic model-based initialization for convolutional neural networks. Journal of Visual Communication and Image Representation. 89:103677. doi:10.1016/J.JVCIR.2022.103677.
[3]. Castillo Camacho Ivan & Wang Kai.(2022).Convolutional neural network initialization approaches for image manipulation detection. Digital Signal Processing. 122:103376. doi:10.1016/J.DSP.2021.103376.
Round 2
Reviewer 1 Report
The implemented additions and changes have significantly improved the indicated research and the presentation of the results of this interesting scientific field.
The quality of the text and the style is correct.
Author Response
Thank you for your comments and suggestions on this article. Thank you very much
Reviewer 2 Report
The paper can be accepted in its present form.
Author Response

(The authors gave the same response as above.)
